# Advancements in Understanding and Classifying Chronic Orofacial Pain: Key Insights from Biopsychosocial Models and International Classifications (ICHD-3, ICD-11, ICOP)

**DOI:** 10.3390/biomedicines11123266

**Published:** 2023-12-09

**Authors:** Federica Canfora, Giulia Ottaviani, Elena Calabria, Giuseppe Pecoraro, Stefania Leuci, Noemi Coppola, Mattia Sansone, Katia Rupel, Matteo Biasotto, Roberto Di Lenarda, Michele Davide Mignogna, Daniela Adamo

**Affiliations:** 1Department of Neuroscience, Reproductive Sciences and Dentistry, University of Naples Federico II, 5 Via Pansini, 80131 Naples, Italy; federica.canfora@unina.it (F.C.); danielaadamo.it@gmail.com (D.A.); 2Department of Surgical, Medical and Health Sciences, University of Trieste, 447 Strada di Fiume, 34149 Trieste, Italy; 3Dentistry Unit, Department of Health Sciences, University of Catanzaro “Magna Graecia”, 88100 Catanzaro, Italy

**Keywords:** orofacial pain, burning mouth syndrome, chronic pain, models, biopsychosocial, toothache/diagnosis, trigeminal neuralgia, International Classification of Diseases (ICD-11), International Classification of Orofacial Pain (ICOP), International Classification of Headache Disorders (ICHD-3), Diagnostic and Statistical Manual of Mental Disorders (DSM-5)

## Abstract

In exploring chronic orofacial pain (COFP), this review highlights its global impact on life quality and critiques current diagnostic systems, including the ICD-11, ICOP, and ICHD-3, for their limitations in addressing COFP’s complexity. Firstly, this study outlines the global burden of chronic pain and the importance of distinguishing between different pain types for effective treatment. It then delves into the specific challenges of diagnosing COFP, emphasizing the need for a more nuanced approach that incorporates the biopsychosocial model. This review critically examines existing classification systems, highlighting their limitations in fully capturing COFP’s multifaceted nature. It advocates for the integration of these systems with the DSM-5’s Somatic Symptom Disorder code, proposing a unified, multidisciplinary diagnostic approach. This recommendation aims to improve chronic pain coding standardization and acknowledge the complex interplay of biological, psychological, and social factors in COFP. In conclusion, here, we highlight the need for a comprehensive, universally applicable classification system for COFP. Such a system would enable accurate diagnosis, streamline treatment strategies, and enhance communication among healthcare professionals. This advancement holds potential for significant contributions to research and patient care in this challenging field, offering a broader perspective for scientists across disciplines.

## 1. Introduction

Chronic pain, affecting an estimated 30% of the global population, ranks as one of the predominant reasons for seeking medical care [1,2]. Despite the higher mortality associated with conditions such as heart attacks, strokes, infectious diseases, cancer, and diabetes, chronic pain remains a leading cause of human suffering and disability, surpassing many other health challenges in its impact on quality of life [1,3,4].

The mismanagement of pain not only leads to severe physical, psychological, and social repercussions but also incurs substantial economic costs, both in terms of healthcare expenditure and lost productivity [5,6,7].

Patients enduring chronic pain often experience a protracted journey through the healthcare system, spanning months or years, without adequate recognition of their condition [8,9]. This lack of validation can contribute to feelings of alienation and exacerbate their suffering [10]. The inability of healthcare systems to effectively acknowledge and address the multifaceted impact of chronic pain has emerged as a critical issue, necessitating urgent attention in medical and dental practices as well as at the policy level [11,12,13].

Understanding and effectively treating chronic pain is not just a clinical challenge but a societal imperative, in medicine, dentistry, and for the national healthcare system [14,15].

In response to the complexities presented by chronic pain, particularly in the orofacial region, this review seeks to illuminate the various dimensions of chronic orofacial pain (COFP) [16,17]. The face and mouth hold special significance in human physiology and social interaction, making pain in this region particularly complex [18,19]. The prevalence of COFP is notable, with approximately 10% of chronic pain cases falling within this category [20,21,22]. It encompasses a range of conditions, from trigeminal neuralgia to burning mouth syndrome, each presenting unique challenges in terms of diagnosis and management [23,24,25]. The review will explore these conditions in depth, focusing on their epidemiology, clinical presentation, and the intricacies of their classification.

A central theme of this review is the exploration of the biopsychosocial model of chronic pain [26,27]. This model emphasizes that pain is not solely a physical symptom but is significantly influenced by psychological and social factors [28,29].

This understanding is crucial for healthcare professionals as it necessitates a highly individualized approach to pain management [30,31,32].

We will specifically examine the classification systems currently in use for COFP, including the International Classification of Orofacial Pain (ICOP 2020), the International Classification of Headache Disorders (ICHD-3), and the International Classification of Diseases (ICD-11). This review will critique these systems, highlighting their strengths and limitations, and discuss the future directions for a more effective and comprehensive classification of COFP. This exploration is vital for enhancing diagnostic accuracy, optimizing treatment modalities, and ultimately improving patient outcomes.

Finally, this review aims to provide a thorough understanding of COFP, its classification, and its management, with a focus on the integration of the biopsychosocial model and the latest diagnostic frameworks, in order to contribute to the broader understanding of orofacial pain within the healthcare community, facilitating better patient care and outcomes.

According to the International Association for the Study of Pain (IASP), “pain is an unpleasant sensory and emotional experience associated with actual or potential tissue damage or described in terms of such damage” [33,34,35].

The IASP task force expanded this definition in 2020 by adding six key notes [35]:

Pain is always a personal experience that is influenced by biological, psychological, and social factors;

Pain and nociception are different phenomena. Pain cannot be solely deduced from the activity of sensory neurons;

Individuals learn the concept of pain through their life experiences;

The individual’s personal history of pain should be respected;

Although pain usually plays an adaptive role, it can have negative effects on social, psychological, and functional well-being;

Verbal description is just one of many behaviors used to express pain. The inability to communicate does not negate the possibility that a human or non-human being experiences pain [35].

Chronic pain is defined as pain that persists beyond the normal time of healing that is associated with a particular type of damage or disease [36]. According to the IASP task force, chronic pain is defined as persistent or recurrent pain lasting more than 3 months [37].

Pain, a complex and multifaceted experience, can be categorized into three distinct types based on mechanistic differences: nociceptive, neuropathic, and nociplastic pain [38].

Nociceptive pain is defined as pain that arises from actual or potential damage to non-neural tissue due to the stimulation of nociceptors [35]. Therefore, neural tissue is generally healthy, and signal transmission is normal; what creates pain is the stimulation of “nociceptors” present both somatically and viscerally [39].

Nociceptors are primary afferent neurons with a high activation threshold. Acute pain is due to the activation of myelinated nociceptors (mainly Aδ fibers), while chronic pain is generally due to the activation of unmyelinated nociceptors [40,41]. This pain type is distinguished from neuropathic pain by its association with a normally functioning somatosensory nervous system [39,42].

Neuropathic pain is defined as pain caused by a lesion or disease affecting the somatosensory nervous system [43]. It is generally perceived within the territory of innervation of the damaged nerve [37].

Depending on the affected system, it is categorized as peripheral or central: it is peripheral when the lesion and disease affect the peripheral system, and it is central when the central nervous system is affected [36,44].

The somatosensory system includes peripheral receptors and neural pathways through which the central nervous system can detect and process information about the body, such as touch, pressure, vibration, pain, temperature, position, and movement of its parts [45,46]. Therefore, by definition, within the concept of pain, a lesion or disease of the somatosensory system is related to altered pain processing [47,48].

Neuropathic pain can be either spontaneous or evoked, marked by either negative (hypoalgesia, hypoesthesia) or positive sensory signs (hyperalgesia, allodynia) that must be perceived in the territory of innervation of the injured nervous structure [49,50] (Appendix A).

Confirming neuropathic pain often requires demonstrating a nervous system lesion or disease via diagnostic methods such as imaging, biopsies, or neurophysiological tests, with questionnaires serving as supplementary screening tools [51].

The concept of nociplastic pain, a term introduced by the pain research community [37,52], represents the third category [53].

Nociplastic pain is characterized by altered nociception in the absence of clear evidence of either actual or potential tissue damage that would activate peripheral nociceptors, or of any injury to the somatosensory system [54]. Typically, this type of pain presents as multifocal, widespread, and intense, and is often accompanied by a constellation of symptoms including fatigue, sleep disturbances, memory impairment, and mood alterations [38].

Baliki et al. showed changes in brain connectivity in patients with nociplastic pain (chronic back pain and osteoarthritis), demonstrating reduced connectivity of the medial prefrontal cortex and increased connectivity of the insular cortex in proportion to the intensity of pain [55].

This type of pain can manifest itself in isolation, as often occurs in conditions such as fibromyalgia or tension-type headache, or as part of a mixed pain state in combination with nociceptive or neuropathic pain in progress, as might occur in chronic low back pain or chronic orofacial pain [56].

It responds differently to treatments, often showing limited response to anti-inflammatory drugs and opioids [38].

It is crucial to recognize that “neuropathic”, “nociceptive”, and “nociplastic” pain are not specific pathologies in themselves but are used to categorize pain types with potentially shared pathophysiological features [57].

The perception and experience of pain are complex processes that have a multidimensional character, influenced by various factors that do not only involve sensory-discriminative biological processes, but also the interaction of affective, motivational, cognitive, and social factors in a biopsychosocial model, as emphasized by the expansion of the definition by the IASP [33,58]. The model considers the interaction of these factors in the experience of pain [58]. This convergence of diverse elements renders everyone’s pain experience uniquely personal, necessitating a highly individualized approach by healthcare professionals [59,60] (Figure 1).

The face and mouth hold a special significance in human physiology and social interaction, serving not just in essential functions like taste, smell, chewing, swallowing, and sensory–motor activities, but also playing a crucial role in interpersonal communication [58,61,62,63]. Consequently, pain in this region gains a multidimensional aspect, often more complex than pain in other body areas [64,65,66]. Facial pain (FP) is commonly defined as pain predominantly occurring below the orbito-meatal line, anterior to the earlobes, and above the neck, with some definitions extending to the forehead, linking certain headaches to orofacial structures [67,68,69,70].

Orofacial pain (OFP) encompasses all structures of the oral cavity [71]. In dental practice, such pain is often associated with dental issues or temporomandibular joint disorders; however, in 30% of cases, dentists may be faced with a type of pain that cannot be attributed to any of the above situations [72,73,74].

Approximately 20% of acute craniofacial pains have the potential to evolve into COFP if not addressed effectively and in good time [16,75]. Dentists play a pivotal role in identifying and managing these conditions, especially when the pain source is not immediately apparent, thus avoiding unnecessary procedures [18,76]. The complexity and often elusive etiology of COFP make diagnosis and treatment challenging, with patients frequently facing delayed diagnosis and intervention, typically 2–3 years post onset, regardless of the pain’s nature [77,78,79].

COFP can be categorized into those with a clear etiology, such as post-herpetic trigeminal neuralgia or post-traumatic neuropathic trigeminal pain [80,81], those associated with chronic diseases like arthritis or diabetes, and a significant number of idiopathic cases where the cause remains unidentified, including conditions like burning mouth syndrome (BMS), persistent idiopathic facial pain (PIFP), or persistent idiopathic dentoalveolar pain (PIDP) [82]. Consequently, COFP is classified as primary when idiopathic and secondary when linked to an identifiable pathological process [16,83]. The impact of COFP extends beyond physical discomfort, often correlating with mood disorders, sleep disturbances, cognitive impairments, and a substantial reduction in quality of life [3,84,85].

Symptomatically, COFP shares common features with other pain types, such as neuropathic characteristics, including spontaneous, intermittent, lancinating, or burning sensations, alongside both positive and negative sensory symptoms [35,86] (Appendix A).

The prevalence of OFP varies, with estimates ranging from 16.1% to 33.2%, and a more realistic figure seems to hover around 25% [87]. Within this spectrum, COFP accounts for about 10%, positioning it among the most common types of chronic pain, after low back, neck, and knee pain [1]. Recent studies indicate an incidence rate of COFP at approximately 38.7 per 100,000 person-years. It is more prevalent in women and tends to increase with age [88].

The majority of COFP cases are linked to musculoskeletal disorders, particularly temporomandibular disorders (TMDs), which affect an average of 4.6% of the general population, with a higher incidence in women (6.3%) compared to men (2.8%) [89].

Another highly prevalent form of COFP is BMS [90].

The worldwide prevalence of BMS is estimated at 1.73% in the general population, which notably increases to 7.72% among dental patients, as detailed in a 2021 meta-analysis conducted by Wu et al. [91]. This extensive study included both clinical and population-based research, offering a comprehensive view. However, the exact prevalence rates vary due to differences in population demographics, geographic regions, and diagnostic criteria used. In terms of geographic distribution, BMS prevalence is higher in Europe (5.58%) and North America (1.10%), compared to Asia (1.05%), within the general population. This trend reverses in clinical settings, with Asia reporting a higher prevalence (8.96%) than South America (6.05%) and Europe (6.46%) [91]. Such variations underscore the significant impact of geographic factors on BMS occurrence. A gender-specific analysis of the data reveals a more pronounced prevalence of BMS in females (1.15%) than in males (0.38%) within the general population, indicating a ratio of approximately 3:1 [92]. This notable gender disparity in BMS prevalence may be linked to physiological and behavioral factors specific to each gender [93].

Trigeminal neuralgia (TN) is relatively rare, with a prevalence ranging from 0.03% to 0.3% in the general population, and an incidence of approximately 1 case per 70,000 to 100,000 individuals annually [94,95]. The likelihood of developing TN increases with age and the presence of certain medical conditions, such as migraines [96]. Notably, TN is 15–20 times more prevalent in individuals with multiple sclerosis, where its prevalence ranges between 1.1% and 6.3% [97].

PIFP is also rare, occurring in about 0.03% of the population, with an incidence rate of 4.4 per 100,000 person-years. PIFP predominantly affects females, typically manifesting around the age of 40 [98].

An ideal classification system in medicine is pivotal, serving as a comprehensive, biologically plausible, clinically useful, and reliable framework. In the realm of OFP, such a system is indispensable for accurate disease identification, effective treatment planning, and enhanced patient understanding and acceptance of their condition.

Classification in medicine organizes pathologies into categories based on specific criteria [99]. For OFP, these criteria often stem from etiology, pathophysiology, diagnosis, and management [72]. Clinicians rely on this classification for precise disease identification and treatment strategies. Patients benefit from a clear diagnosis framed within a globally recognized system, aiding in their understanding and acceptance of the disease. Moreover, for research purposes, classification is crucial as it ensures consistent diagnostic criteria are applied across studies, whether in pharmacological trials or pathophysiological investigations.

According to Fillingim et al. [100], an ideal classification system should be:exhaustive (including all clinical diseases or disorders within the field of interest);biologically plausible (symptoms and signs should correspond to known biological processes) and mutually exclusive (there should be no overlap between disease entities due to common symptoms);clinically useful (so that it can be used to aid in treatment and prognosis);reliable (applicable consistently and reproducibly among clinicians);simple for practical use.

However, most current OFP classification systems suffer from deficits in at least one of these qualities.

The variability in definitions across different classifications, drafted by independent clinician groups, further complicates this issue.

Although there is no consensus yet regarding a universal and unique OFP classification, the latest ICOP 2020 classification [71] seems to encompass most of Fillingim et al.’s requirements [100].

Other notable systems include:The International Classification of Headache Disorders, 3rd edition (ICHD-3) beta version [101];The International Association for the Study of Pain for the International Classification of Diseases (ICD-11) [14].

The ongoing updates to these classifications, readily available online, reflect the scientific community’s commitment to enhancing the understanding of these pathologies. This process exemplifies the dynamic and evolving nature of medical knowledge, driven by continual research and clinical findings. As new insights emerge, classification systems are systematically refined and expanded, ensuring they remain current and applicable. This iterative approach is integral to advancing medical science and improving diagnostic accuracy, ultimately leading to better patient outcomes and more effective treatments.

The International Classification of Headache Disorders, 3rd edition (ICHD-3) beta version.

Since its inaugural publication in 1988 by the International Headache Society (IHS), the ICHD has undergone significant revisions, with major updates in 2004, 2013 (ICHD-3), in 2018 (ICHD-3 beta version), and in 2020 [67,101]. This taxonomic classification system, primarily focused on headache-related disorders and painful craniofacial conditions, is organized into a three-part system. Parts I and II categorize primary and secondary headaches across 12 main groups. Part III addresses painful cranial neuropathies and other facial pain conditions, particularly those related to the trigeminal nerve [102] (Table 1).

This part has two distinct sections, one for primary painful craniofacial neuralgias and facial pain and the other for unclassified or unspecified headaches. A reference to OFP conditions also appears in Part II of the ICHD-3, dedicated to secondary headaches, which occur when a primary headache worsens in close temporal relationship with a causative OFP condition. Therefore, the ICHD-3 recognizes that some disorders arising from dental, maxillary, maxillary–sinus, eye, ear, and nose elements can cause headaches. However, despite its comprehensive structure, ICHD-3 falls short in encompassing all clinical phenotypes of OFP and does not adequately consider biopsychosocial and behavioral factors [102,103].

The hierarchical nature of the ICHD-3 allows for varying levels of diagnostic detail, adaptable to different clinical settings in which clinicians may decide the necessary level of diagnosis based on the type of practice. Therefore, coding can range from one digit up to five depending on the detail of the criteria used in the diagnosis. This flexibility is crucial for accurately diagnosing and coding over 200 headache-related disorders categorized in 12 main headache categories. The ultimate goal is to align this coding system with the WHO’s International Classification of Diseases (ICD-11) [14,33].

In oral medicine, the most frequently referenced category within Part III is Category 13, dealing specifically with OFP conditions [104]. (Table 1)

Parallel to the ICHD’s development, the IASP has collaborated with the WHO to integrate a systematic classification of chronic pain into the ICD-11.

Every update or recent development related to the ICHD or ICD-11 since 2018 is available on-line.

This integration represents a significant step towards a more comprehensive understanding of chronic pain conditions, including OFP, and emphasizes the need for a multidimensional approach to diagnosis and treatment.

The International Classification of Diseases 11th Revision (ICD-11).

The 11th Revision of the ICD-11 [12], published by the WHO in June 2018, represents a significant evolution in the global standard for health data classification. Its international and multilingual taxonomy is used for collecting and reporting data on health services (e.g., statistics on morbidity and mortality, quality and safety, health costs, and clinical research).

This latest revision, which has been the international standard for coding since 2022, offers enhanced features for research applicability, clinical detail encoding, and modernized scientific content. One of the pivotal advancements in ICD-11, as highlighted by Scholz et al. [37], is its refined approach to chronic pain, now recognized both as a symptom and as a distinct disease entity, complete with a supportive coding framework.

A groundbreaking feature of ICD-11 is the concept of “multiple parenting”, which allows diagnostic entities to be classified under more than one category. This flexibility overcomes previous limitations where diseases could only be classified singularly by etiology or site [14]. For instance, “Chemotherapy-induced chronic pain” can be categorized both under “Chronic cancer pain” (etiology) and “Chronic neuropathic pain” (mechanism). This multifaceted approach aligns more closely with the interdisciplinary nature of medical practice, bridging gaps between specialties such as oncology and neurology.

The ICD-11 pain classification introduces nine main diagnostic categories [14] (Table 2).

Specifically, neuropathic pain conditions are divided into two major categories:Chronic central neuropathic pain (MG 30.50);Chronic peripheral neuropathic pain (MG 30.51).

Orofacial pain in the ICD-11 is recognized as a significant medical entity, reflecting advancements in understanding pain as a complex, multidimensional experience [105].

Notably, OFP is categorized under secondary chronic headaches (code MG 30.62) with specific codes assigned to different conditions such as NT, PIFP, and BMS [14,106].

Specific codes for orofacial pain conditions:NT: 8B82.0 under trigeminal disorders (Foundation ID in the ICD-11 browser: 1803581281);PIFP: 8B82.1 under trigeminal disorders (Foundation ID in the ICD-11 browser: 248232693);BMS: DA0F.0: sensory disturbances involving the orofacial region (Foundation ID in the ICD-11 browser: 618998878) (Figure 2).

See Appendix B for extension codes.

See website https://icd.who.int/en accessed on 5 November 2023 for the complete classification of ICD-11.

The codes for the most common forms of OFP in different classifications are indicated in Table 3.

The ICOP 2020 marks a significant evolution in the understanding and classification of OFP [71]. This system, modeled after the ICHD-3, offers a nuanced approach to diagnosing OFP based on symptomatology and suspected pathophysiology, prioritizing the disorder’s characteristics over its location. This novel approach broadens the diagnostic criteria and fosters a more comprehensive understanding of various OFP forms improving a proper diagnosis and management.

The hierarchical classification system of the ICOP 2020 enables a granular diagnosis, ranging from a broad categorization (first digit) to highly specific classifications (up to the seventh digit) [71]. This level of detail adapts to different clinical contexts, with general medicine often utilizing first or second digit diagnoses, while specialist practices may require more detailed coding.

Unlike ICHD-3, primary and secondary OFP are included in the same group in separate subgroups because, according to experts, the strict structure of the ICHD criteria, using Criterion A to describe the pain, Criterion B to identify the presumed cause, and Criterion C to establish evidence of causality, is not easily applicable to OFP.

In cases where patients present with multiple types of pain, diagnoses should be prioritized based on the patient’s experience of suffering and disability, also including the longitudinal history of pain (how and when did the pain start?), family history, medication effect, relationship with menstrual cycle, age, sex, etc.

For a definitive OFP diagnosis, patients must experience a minimum number of attacks or days with pain, as specified in the detailed criteria for each type or subtype of OFP. Different OFPs must meet a series of other requirements described in the criteria under separate letter headings, i.e., A, B, C, etc. Some letter headings are monothematic, expressing a single requirement. Other letter headings are polythematic and require, for example, two of the four listed characteristics. This structure is common in the ICHD-3 [67].

The diagnostic criteria in this classification do not include the evaluation of the severity and frequency of painful attacks, which should be evaluated and specified [107,108].

Another innovative feature of this classification is the introduction of the “psychosocial assessment of patients with OFP”. This approach integrates psychometric testing for a holistic evaluation of the patient, acknowledging the significant role of psychological factors in pain perception and management [71]. The recommended tests are summarized in Table 4. The ICOP classification considers six groups. However, in this text, we will focus on pain types 4 and 6 (Table 5). (See cross-references for the complete classification of ICOP 2020).

## 2. Discussion and Future Directions

The modern approach to understanding COFP emphasizes the importance of a comprehensive biopsychosocial perspective. This model holistically considers the intertwined nature of biological (physiological), psychological, and social factors in pain perception and management. By advocating for the integration of diagnostic codes from various contemporary classification systems—the ICOP 2020, the ICHD-3, and the ICD-11—a more nuanced understanding of COFP is fostered. Additionally, incorporating the code for somatic symptom disorder (SSD) with predominant pain from the Diagnostic and Statistical Manual of Mental Disorders, Fifth Edition (DSM-5), by the American Psychiatric Association, underscores the crucial role of psychological factors in the experience of chronic pain. This integration acknowledges that COFP is not solely a physical ailment but a complex interplay of various aspects of human health, necessitating a multifaceted approach to diagnosis and treatment [14,71,103,109]. This integration, aligned with the ICD coding system, could enhance the standardized coding of chronic pain, including in mental health contexts, underscoring the vital interplay of biological, psychological, and social factors in chronic pain syndromes [12,110].

The primary DSM-5 code for somatic symptom disorder is [109]—300.82 (F45.1)—somatic symptom disorder.

In the realm of COFP, and other chronic pain types, this disorder can be conceptualized within the SSD framework when patients exhibit heightened attention to physical symptoms such as pain, accompanied by intense health-related anxiety, disproportionate concerns regarding their symptoms, and significant investment of time and energy in these concerns, leading to considerable distress or functional impairment [111,112].

These persistent reactions often continue even after extensive medical evaluations and reassurances. The DSM-5’s transition from the DSM-4 has shifted the focus from the presence of physical complaints to the subjective experience of distress [113,114]. The DSM-5 criteria for SSD with predominant pain emphasize the excessive thoughts, feelings, and behaviors related to somatic symptoms, irrespective of whether the pain has a known medical cause [115,116,117].

In instances where COFP patients display characteristics indicative of SSD, this diagnostic code might be considered supplementary. This alignment with modern pain classification systems underscores the importance of a multidisciplinary and holistic approach, centering patient-focused care and treatment strategies [118,119,120] (Figure 3).

This review, comprising an in-depth examination of COFP and its classification systems, sets the stage for a comprehensive understanding of this complex condition [121,122]. Beginning with a summary of the previous sections, we have explored the prevalence and impact of chronic pain, the intricacies of COFP, and existing classification frameworks such as the ICOP 2020, ICHD-3, and ICD-11. In this study, we delve into the potential for extension of previous works on COFP, highlighting its implications and paving the way for future research. By providing a holistic view of COFP classification and management, this study significantly extends the current understanding of the condition. It emphasizes the integration of the biopsychosocial model, underlining the importance of considering not just the biological aspects of pain, but also incorporating psychological and social factors into our perception and treatment of pain. This comprehensive approach marks a pivotal advancement in the field, offering new directions for subsequent investigations [123,124,125]. This approach offers a significant extension of traditional perspectives, which predominantly focused on the physiological aspects of pain [126].

The implications of this review are far-reaching. It challenges the existing paradigms in COFP diagnosis and management, advocating for a more integrated approach [16,127,128]. The review highlights the limitations of current classification systems and suggests that a more nuanced system is essential for better patient outcomes. This could significantly influence how healthcare providers approach COFP, leading to more personalized and effective treatment plans [129,130].

This review lays the groundwork for future research in several ways. Firstly, by identifying gaps in current classification systems, it calls for the development of a unified framework that can adapt to evolving understandings of COFP [131,132]. Secondly, it opens avenues for research into the integration of psychological and social factors in pain management, which is relatively unexplored in COFP. This could lead to new treatment modalities that address the patient’s overall well-being [133,134,135].

The ultimate goal is to establish a comprehensive, universally applicable classification system for COFP that effectively integrates the biopsychosocial model. The challenge lies in balancing the need for a system that is detailed enough to be clinically useful yet simple enough for widespread adoption. Additionally, overcoming the traditional siloed approach to pain management and fostering a more holistic, interdisciplinary methodology remains a significant hurdle [136,137].

Achieving this goal requires an amalgamation of knowledge from various fields, including neurology, psychology, and sociology, coupled with advancements in technology [138]. Machine learning and data analytics could play a crucial role in analyzing patient data to derive more precise classifications [139]. Additionally, developing technologies for better pain measurement and monitoring could significantly enhance diagnosis and treatment [140,141].

The field of research focusing on COFP holds significant importance, extending far beyond the boundaries of a singular medical condition. This line of inquiry is instrumental in enhancing the overall understanding of COFP, a complex and often debilitating condition that impacts a substantial portion of the population. By delving into the multifaceted aspects of COFP, including its origins, progression, and effective management strategies, this research contributes profoundly to the broader discipline of pain management. It offers invaluable insights into the nuanced interplay between physiological, psychological, and social factors in pain perception and response. Furthermore, advancements in COFP research have the potential to inform and improve therapeutic approaches, leading to more effective and personalized treatment plans for various types of pain conditions. This not only benefits patients suffering from COFP but also enriches the entire field of pain management with new knowledge and perspectives, ultimately enhancing the quality of life for individuals affected by chronic pain worldwide [142]. It challenges the existing frameworks and methodologies, pushing for innovations that can have a transformative impact on healthcare practices [143,144].

Despite its comprehensive approach, this review has limitations. The primary limitation is the reliance on existing literature, which may not fully capture the latest developments in the field.

However, the strengths of this study include a comprehensive examination of the various types of COFP, their underlying mechanisms, and their impact on patients’ quality of life through an in-depth analysis of current classification systems [145,146]. However, the manuscript also emphasizes the significance of using the biopsychosocial model to understand pain perception and integrating biological, psychological, and social dimensions in pain management, which provide valuable insights for both clinicians and researchers [147].

Clinically, this review can aid in the development of more effective treatment strategies for COFP, leading to better patient outcomes and quality of life, taking into account the importance of psychometric evaluation in the treatment of patients with COFP as well as the importance of evaluating psychiatric comorbidities when making treatment decisions [148,149].

Finally, this review marks a significant step forward in the understanding of COFP. By highlighting the limitations of the current systems and advocating for a more holistic approach, it sets the stage for future advancements in this field. The integration of diverse knowledge domains and the application of new technologies are key to realizing the potential of this research, ultimately contributing to the evolution of pain management practices [144,150].

## 3. Conclusions

This review advocates for an integrated, multidisciplinary approach to COFP, emphasizing the necessity of considering physical, psychological, and social factors in pain management. The adoption of the biopsychosocial model is essential in developing more effective treatments that focus on a patient’s overall well-being. Our analysis of classification systems like the ICOP 2020, ICHD-3, and ICD-11 underscores the need for a refined, universally applicable diagnostic framework that aligns with the evolving understanding of COFP. We highlight the challenges in accurate diagnosis and classification, indicating the need for continuous methodological and theoretical advancement. Future research should aim at creating a unified classification system capturing COFP’s complexity and investigating how psychological and social factors can be integrated into individualized treatment plans. This work signals a shift towards a more holistic, patient-centered approach to COFP in both clinical and research settings.

## Figures and Tables

**Figure 1 biomedicines-11-03266-f001:**
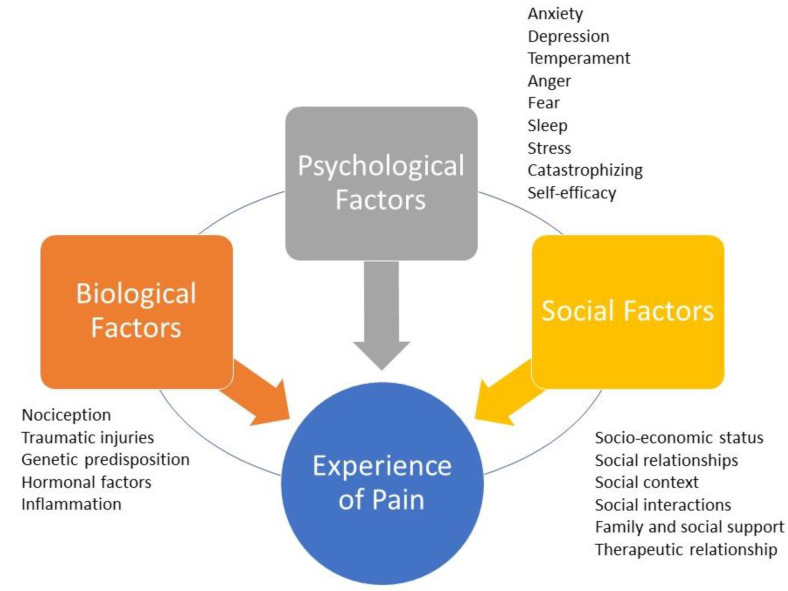
The biopsychosocial model of pain. The experience of pain is personal because it is mediated by the dynamic, interdependent, and synergistic integration of three factors/domains: biological factors, psychological factors, and social factors. Each domain may have characteristics shared with the others. Biological factors that can negatively affect the perception of pain include traumatic injury, the severity of overall health conditions, and genetic/hormonal predisposition. Psychological factors that can amplify pain include anxiety, depression, sleep disorders, excessive catastrophizing, and low self-efficacy. Social factors that can negatively influence the perception of pain include gender roles, ethnic identity, discrimination, poor family support, and healthcare provider prejudice.

**Figure 2 biomedicines-11-03266-f002:**
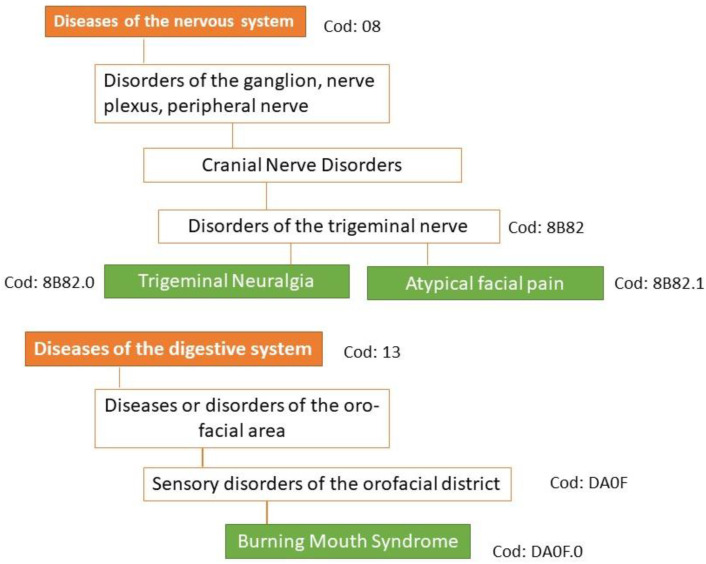
Classification of the three main types of orofacial pain according to ICD-11. Under the new concept of multiple parenting in ICD-11, each entity can be assigned to other divisions of the classification of chronic pain. In ICD-11, persistent idiopathic facial pain is referred to as atypical facial pain.

**Figure 3 biomedicines-11-03266-f003:**
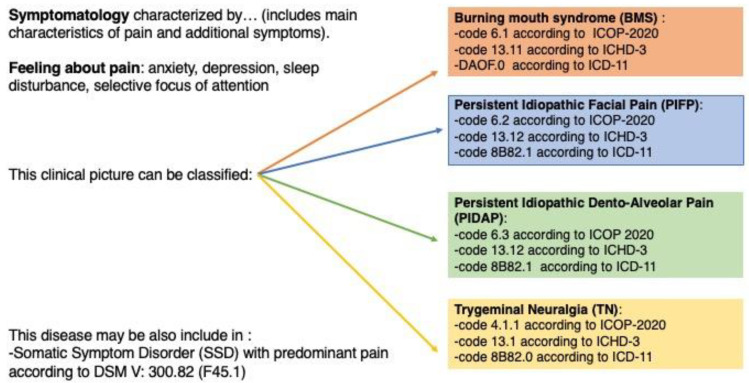
Classification and coding of chronic orofacial pain disorders. Abbreviations—BMS: burning mouth syndrome; PIFP: persistent idiopathic facial Pain; PIDP: persistent idiopathic dento-alveolar pain; TN: trigeminal neuralgia; SSD: somatic symptom disorder; ICOP: International Classification of Orofacial Pain (2020 Edition); ICHD-3: International Classification of Headache Disorders (3rd Edition); ICD-11: International Classification of Diseases (11th Revision); DSM-5: Diagnostic and Statistical Manual of Mental Disorders (5th Edition).

**Table 1 biomedicines-11-03266-t001:** Classification system from the International Headache Society (ICHD-3 beta version).

Part 1: Primary Headache	
1. Migraine	
2. Tension-type headache	
3. Trigeminal autonomic cephalalgias	
4. Other primary headache disorders	
PART 2: SECONDARY HEADACHE	
5. Headache attributed to trauma or injury to the head and/or neck	
6. Headache attributed to cranial or cervical vascular disorder	
7. Headache attributed to non-vascular intracranial disorder	
8. Headache attributed to a substance or its withdrawal	
9. Headache attributed to infection	
10. Headache attributed to homeostasis disorders	
11. Headache or facial pain attributed to disorders of the skull, neck, eyes, ears, nose, paranasal sinuses, teeth, mouth, or other facial or cervical structures	
12. Headache attributed to psychiatric disorder	
PART 3: CRANIAL NEUROPATHIES, AND OTHER FACIAL PAINS	
13. Painful cranial neuropathies and other facial pains	
14. Other headache disorders	
13. Painful cranial neuropathies and other facial pains	
ICHD code	Types of Pain
13.1	Trigeminal neuralgia
13.1.1	Classical trigeminal neuralgia
13.1.1.1	Purely paroxysmal classical trigeminal neuralgia
13.1.1.2	Classical trigeminal neuralgia with concurrent persistent facial pain
13.1.2	Painful trigeminal neuropathy
13.1.2.1	Painful trigeminal neuropathy attributed to acute herpes zoster
13.1.2.2	Post-herpetic trigeminal neuropathy
13.1.2.3	Painful post-traumatic trigeminal neuropathy
13.1.2.4	Painful trigeminal neuropathy attributed to multiple sclerosis (MS) plaques
13.1.2.5	Painful trigeminal neuropathy attributed to space-occupying lesion
13.1.2.6	Painful trigeminal neuropathy attributed to other disorders
13.11	Burning mouth syndrome (BMS)
13.12	Persistent idiopathic facial pain (PIFP)

Abbreviations: OFP: orofacial pain; ICOP: International Classification of Orofacial Pain; IASP: International Association for the Study of Pain; ICD: International Classification of Diseases. Trigeminal neuralgia attributed to multiple sclerosis and trigeminal neuralgia attributed to space-occupying lesion are only coded within the context of central chronic neuropathic pain (MG 30.50).

**Table 2 biomedicines-11-03266-t002:** International Association for the Study of Pain’s (IASP) classification system with the International Classification of Diseases (ICD-11), categorizing different types of chronic pain.

Types of Pain	ICD-11 Code
Primary chronic pain	MG 30.0
Chronic cancer pain	MG 30.1
Chronic post-surgical and post-traumatic pain	MG 30.2
Chronic musculoskeletal pain	MG 30.3
Secondary chronic visceral pain	MG 30.4
Chronic neuropathic pain	MG 30.5
Secondary chronic headache or orofacial pain	MG 30.6
Other specified chronic neuropathic pain	MG 30.Y
Chronic neuropathic pain, unspecified	MG 30.Z

**Table 3 biomedicines-11-03266-t003:** Codes for common forms of orofacial pain according to different classification systems.

Types of OFP	ICOP 2020	ICHD-3 v. Beta	IASP for ICD-11
Trigeminal neuralgia	4.1.1	13.1	8B82.0
Purely paroxysmal classical trigeminal neuralgia	4.1.1.1.1	13.1.1.1	N.D
Classical trigeminal neuralgia with concomitant continuous pain	4.1.1.1.2	13.1.1.2	N.D
Trigeminal neuralgia attributed to multiple sclerosis	4.1.1.2.1	13.1.2.4	MG 30.50
Trigeminal neuralgia attributed to space-occupying lesion	4.1.1.2.2	13.1.2.5	MG 30.50
Trigeminal neuropathic pain attributed to herpes zoster	4.1.2.1	13.1.2.1	1E91.4
Post-herpetic trigeminal neuralgia	4.1.2.2	13.1.2.2	1E91.5
Post-traumatic trigeminal neuropathic pain	4.1.2.3	13.1.2.3	N.D
Trigeminal neuralgia attributed to another cause	4.1.1.2.3	13.1.2.6	8B82.Z
Burning mouth syndrome	6.1	13.11	DA0F.0
Persistent idiopathic facial pain	6.2	13.12	8B82.1
Persistent idiopathic dento-alveolar pain	6.3	N.D	N.D

**Table 4 biomedicines-11-03266-t004:** Recommended psychometric tests for the assessment of pain, psychological profile, and coping strategies of the patient.

Abbreviations	Questionnaires	
Pain		
Pain drawing	/	Evaluate the extent of the pain
GCPS	Graded Chronic Pain Scale	Scale of pain intensity and disability
JFLS	Jaw Functional Limitation Scale	Scale of functional limitation (specific to TMD)
OBC	Oral Behaviors Checklist	Checklist of dysfunctional behaviors (specific to TMD)
Depression and anxiety		
GAD7	General Anxiety Disorders	Scale for the evaluation of generalized anxiety disorder
PHQ-9	Patient Health Questionnaire-9	Depression evaluation
PHQ-15	Patient Health Questionnaire-15	Scale for the evaluation of the severity of somatic symptoms
PCS	Pain Catastrophizing Scale	Scale for the evaluation of catastrophizing
Coping strategies		
CSQ	Coping Strategies Questionnaires	Scale for the evaluation of coping strategies
TSK	Tampa Scale for Kinesiophobia	Scale of pain avoidance

**Table 5 biomedicines-11-03266-t005:** Classification system for COFP by the International Classification of Orofacial Pain (ICOP 2020).

Orofacial Pain	
1. Orofacial pain attributed to disorders of the dento-alveolar and anatomically related structures	
2. Myofascial orofacial pain	
3. Temporomandibular joint (TMJ) pain	
4. Orofacial pain attributed to lesion or disease of cranial nerves	
5. Orofacial pains with presentations similar to primary headaches	
6. Idiopathic orofacial pain	
**ICOP Code**	**Types of Pain**
4.1	Pain attributed to lesion or disease of the trigeminal nerve
4.1.1	Trigeminal neuralgia
4.1.1.1	Classical trigeminal neuralgia
4.1.1.1.1	Purely paroxysmal classical trigeminal neuralgia
4.1.1.1.2	Classical trigeminal neuralgia with concomitant continuous pain
4.1.1.2	Secondary trigeminal neuralgia
4.1.1.2.1	Trigeminal neuralgia attributed to multiple sclerosis
4.1.1.2.2	Trigeminal neuralgia attributed to space-occupying lesion
4.1.1.2.3	Trigeminal neuralgia attributed to another cause
4.1.1.3	Idiopathic trigeminal neuralgia
4.1.1.3.1	Idiopathic trigeminal neuralgia, purely paroxysmal
4.1.1.3.2	Idiopathic trigeminal neuralgia with concomitant continuous pain
4.1.2	Other trigeminal neuropathic pain
4.1.2.1	Trigeminal neuropathic pain attributed to herpes zoster
4.1.2.2	Post-herpetic trigeminal neuralgia
4.1.2.3	Post-traumatic trigeminal neuropathic pain
4.1.2.3.1	Probable post-traumatic trigeminal neuropathic pain
4.1.2.4	Trigeminal neuropathic pain attributed to another disorder
4.1.2.4.1	Probable trigeminal neuropathic pain attributed to another disorder
4.1.2.5	Idiopathic trigeminal neuropathic pain
**ICOP Code**	**Types of Pain**
6.1	Burning mouth syndrome (BMS)
6.1.1	Burning mouth syndrome without somatosensory alterations
6.1.2	Burning mouth syndrome with somatosensory alterations
6.1.3	Probable burning mouth syndrome
6.2	Persistent idiopathic facial pain (PIFP)
6.2.1	Persistent idiopathic facial pain without somatosensory alterations
6.2.2	Persistent idiopathic facial pain with somatosensory alterations
6.2.3	Probable persistent idiopathic facial pain
6.3	Persistent idiopathic dento-alveolar pain (PIDP)
6.3.1	Persistent idiopathic dento-alveolar pain without somatosensory alterations
6.3.2	Persistent idiopathic dento-alveolar pain with somatosensory alterations
6.3.3	Probable persistent idiopathic dento-alveolar pain

## Data Availability

The data that support the findings of this study are available from the corresponding author, G.O., upon reasonable request.

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
