# Peer review of "Advancements in Understanding and Classifying Chronic Orofacial Pain: Key Insights from Biopsychosocial Models and International Classifications (ICHD-3, ICD-11, ICOP)"

_biomedicines, 2023, doi:10.3390/biomedicines11123266_

Round 1
Reviewer 1 Report
Comments and Suggestions for Authors
The authors have written a comprehensive review of the complexities of chronic orofacial pain. This review is up to date and well written. Of particular use to clinicians and researchers, the authors have included pertinent details regarding the latest classification systems, including the ICOP 2020, ICHD-3 and ICD-11. There are no major omissions. I consider that this review will be of interest to a range of practitioners and researchers in medicine, neurology, ENT surgery and dental medicine.
Comments on the Quality of English LanguageOccasionally, the authors use an apostrophe-s contraction, for example in line 249. While their use of the possessive apostrophe is grammatically correct, perhaps a more formal rendition would be more appropriate in this paper. Thus, in line 249, it might be better to write that "The hierarchical classification system of the ICOP 2020 enables a granular diagnosis ..."
Author Response
We appreciate your feedback on the use of apostrophe-s contractions in our paper. In line with your suggestion for a more formal tone, we have revised line 249 to reflect this. The sentence now reads, "The hierarchical classification system of the ICOP 2020 enables a granular diagnosis..." This adjustment aligns with the formal style appropriate for the context of our paper. Thank you for your attentive and valuable input.

Reviewer 2 Report
Comments and Suggestions for Authors
Canfora, et al. discuss the classification systems for the coding of chronic orofacial pain. I have a few suggestions that I believe will improve the manuscript:
· The introduction should be shortened to focus more explicitly on chronic orofacial pain and written in a more cohesive style. Currently, it is very choppy – as if the information is being presented as bullet points. Also, much of it focuses on chronic pain in general, rather than focusing specifically on chronic orofacial pain. A sentence describing the scope/purpose of the review should be added at the end of the introduction.
· I suggest that the discussion of the classification systems be reordered to ICHD, ICD-11, then ICOP. This would allow the authors to first highlight the limitations of the ICHD and ICD-11, and then describe how the ICOP overcomes some of these limitations.
· Table 1 should be deleted. While it may be a useful glossary, defining these terms is tangential to the main focus of the article.
· Figure 3 includes a notation of “Mr./Mrs.”, but it is unclear what this means in the context of the figure. Please correct.
Author Response
- The introduction should be shortened to focus more explicitly on chronic orofacial pain and written in a more cohesive style. Currently, it is very choppy – as if the information is being presented as bullet points. Also, much of it focuses on chronic pain in general, rather than focusing specifically on chronic orofacial pain. A sentence describing the scope/purpose of the review should be added at the end of the introduction.
Thank you for your constructive feedback regarding the introduction of our manuscript, specifically focusing on chronic orofacial pain. In response to your comments, we have undertaken a thorough revision of the introduction section. Our aim was to make it more concise and directly relevant to chronic orofacial pain, while also enhancing the cohesiveness of the writing style. We took special care to shift the focus from general chronic pain to specifically address chronic orofacial pain, in line with the main theme of our paper. Additionally, we have added a sentence at the end of the introduction to clearly outline the scope and purpose of the review. This revision also aligns with the suggestions made by the Academic Editor, ensuring that the introduction now provides a clearer, more focused, and cohesive overview of the topic. We believe these modifications significantly improve the flow and relevance of the introduction, and we are grateful for the guidance that helped refine our manuscript.
- I suggest that the discussion of the classification systems be reordered to ICHD, ICD-11, then ICOP. This would allow the authors to first highlight the limitations of the ICHD and ICD-11, and then describe how the ICOP overcomes some of these limitations.
Thank you for your insightful suggestion regarding the order of discussion for the classification systems in our manuscript. Based on your recommendation, we have restructured this section to first discuss the ICHD and the ICD-11. This reordering allows us to effectively highlight the limitations inherent in the ICHD and ICD-11 systems. This revised sequence enables us to more clearly demonstrate how the ICOP addresses and potentially overcomes some of the limitations identified in the ICHD and ICD-11 systems. We believe that this reorganization not only improves the logical flow of the discussion but also enhances the clarity of our argument regarding the evolution and improvement of these classification systems. We appreciate your valuable input, which has significantly contributed to the refinement of our manuscript.
- Table 1 should be deleted. While it may be a useful glossary, defining these terms is tangential to the main focus of the article.
Thank you for your suggestion regarding Table 1 in our article. We agree that while the information in Table 1 is useful as a glossary, it might not directly align with the main focus of the text. Consequently, we have relocated Table 1 to the appendix section to maintain its utility without detracting from the article's primary narrative. Additionally, we have renumbered all subsequent tables in the main text to ensure consistency and ease of reference. We believe this adjustment effectively addresses your concerns while preserving the informational value of Table 1.
- Figure 3 includes a notation of “Mr./Mrs.”, but it is unclear what this means in the context of the figure. Please correct.
Thank you for bringing to our attention the issue with the notation of “Mr./Mrs.” in Figure 3 of our article. Upon reviewing your comment, we recognized that this notation might indeed be ambiguous and potentially confusing in the context of the figure. To address this, we have carefully revised Figure 3 and removed the “Mr./Mrs.” notation to enhance clarity and ensure that the figure more accurately conveys the intended information. We appreciate your meticulous observation and hope that this amendment aligns with the overall clarity and quality of the manuscript.

Round 2
Reviewer 2 Report
Comments and Suggestions for Authors
Thank you for the time and effort you took in addressing my comments. I have no further suggestions.
Author Response
Thank you for your acknowledgment and feedback. We greatly appreciate the guidance you have provided throughout this process, which has been instrumental in enhancing the quality of our work.